# Glycogen Synthase Kinase-3β Facilitates Cytokine Production in 12-O-Tetradecanoylphorbol-13-Acetate/Ionomycin-Activated Human CD4^+^ T Lymphocytes

**DOI:** 10.3390/cells9061424

**Published:** 2020-06-08

**Authors:** Cheng-Chieh Tsai, Chin-Kun Tsai, Po-Chun Tseng, Chiou-Feng Lin, Chia-Ling Chen

**Affiliations:** 1Department of Nursing, Chung Hwa University of Medical Technology, Tainan 703, Taiwan; cctsai@mail.hwai.edu.tw; 2Department of Long Term Care Management, Chung Hwa University of Medical Technology, Tainan 703, Taiwan; 3Department of Microbiology and Immunology, National Cheng Kung University, Tainan 701, Taiwan; S46991091@mail.ncku.edu.tw; 4Department of Microbiology and Immunology, School of Medicine, College of Medicine, Taipei Medical University, Taipei 110, Taiwan; iluc0720@hotmail.com (P.-C.T.); cflin2014@tmu.edu.tw (C.-F.L.); 5Core Laboratory of Immune Monitoring, Office of Research & Development, Taipei Medical University, Taipei 110, Taiwan; 6Graduate Institute of Medical Sciences, College of Medicine, Taipei Medical University, Taipei 110, Taiwan; 7School of Respiratory Therapy, College of Medicine, Taipei Medical University, Taipei 110, Taiwan; 8Pulmonary Research Center, Wan Fang Hospital, Taipei Medical University, Taipei 116, Taiwan

**Keywords:** glycogen synthase kinase-3, cytokine, 12-O-tetradecanoylphorbol-13-acetate/ionomycin, CD4^+^ T lymphocyte

## Abstract

Cytokines are the major immune regulators secreted from activated CD4^+^ T lymphocytes that activate adaptive immunity to eradicate nonself cells, including pathogens, tumors, and allografts. The regulation of glycogen synthase kinase (GSK)-3β, a serine/threonine kinase, controls cytokine production by regulating transcription factors. The artificial in vitro activation of CD4^+^ T lymphocytes by a combination of 12-O-tetradecanoylphorbol-13-acetate and ionomycin, the so-called T/I model, led to an inducible production of cytokines, such as interferon-γ, tumor necrosis factor-α, and interleukin-2. As demonstrated by the approaches of pharmacological targeting and genetic knockdown of GSK-3β, T/I treatment effectively caused GSK-3β activation followed by GSK-3β-regulated cytokine production. In contrast, pharmacological inhibition of the proline-rich tyrosine kinase 2 and calcineurin signaling pathways blocked cytokine production, probably by deactivating GSK-3β. The blockade of GSK-3β led to the inhibition of the nuclear translocation of T-bet, a vital transcription factor of T lymphocyte cytokines. In a mouse model, treatment with the GSK-3β inhibitor 6-bromoindirubin-3’-oxime significantly inhibited T/I-induced mortality and serum cytokine levels. In summary, targeting GSK-3β effectively inhibits CD4^+^ T lymphocyte activation and cytokine production.

## 1. Introduction

T lymphocytes play essential roles in controlling adaptive immunity by eliminating infectious pathogens and tumors [1]. Under abnormal conditions, activated T lymphocytes may be involved in immunological diseases, mainly by oversecreting cytokines/chemokines and cytotoxic factors [2]. In cell-mediated immunity, major histocompatibility complex (MHC) class I-restricted CD8^+^ T lymphocytes, also known as cytotoxic T lymphocytes, directly kill their target cells by apoptosis via perforins, granzymes, and Fas (CD95) ligand. In addition, MHC class II-restricted CD4^+^ T lymphocytes, referred to as helper T (Th) lymphocytes, assist innate and adaptive immune responses mainly by producing cytokines/chemokines [3]. Once CD4^+^ T lymphocytes are activated, many cytokines induce the differentiation of naive Th cells into at least five distinct subsets of CD4^+^ effector T cells, namely, Th1, Th2, Th17, follicular helper T, and induced regulatory T (iTreg) cells, on the basis of the production and function of cytokines [4]. For Th1 cell differentiation, signal transducer and activator of transcription (STAT) 4 and STAT1 are activated by interleukin (IL)-12 and interferon (IFN)-γ. During differentiation, activated STAT1 induces the expression of T box transcription factor (T-bet), a transcription factor that initiates IFN-γ, TNF-α, and IL-2 gene expression to promote cell-mediated immunity [5].

For T lymphocyte stimulation, the MHC-peptide complex of antigen-presenting cells causes T cell receptor (TCR) activation and then initiates TCR signal transduction [6,7,8]. An SRC family protein tyrosine kinase, Lck, is associated with the CD4 and CD8 coreceptors, and coclustering with TCR promotes Lck-mediated phosphorylation of immunoreceptor tyrosine-based activation motifs in the CD3 chains, enabling the docking and activation of the protein tyrosine kinase Zeta chain of T cell receptor associated protein kinase (ZAP) 70 [7]. Activated ZAP70, in turn, activates linkers that allow the activation of T cells to promote phospholipase C-γ (PLC-γ) activation. Then, active PLC-γ hydrolyzes phosphatidylinositol bisphosphate to yield inositol trisphosphate and diacylglycerol to trigger an intracellular calcium influx and protein kinase C activation, respectively [6]. This calcium influx activates the phosphatase calcineurin, known as protein phosphatase (PP) 2B, and then stimulates the activation of the nuclear factor activated T cells (NFAT) [6]. Protein kinase C (PKC) signaling is required to activate several signaling pathways, including the nuclear factor-κB (NF-κB) and mitogen-activated protein kinase-regulated activator protein (AP) 1 pathways. Through the interregulation of NFAT, NF-κB, AP-1, and T-bet, all T lymphocyte-associated signaling pathways are important for T lymphocyte activation, proliferation, differentiation, and cytokine production [5,6,7,8].

Glycogen synthase kinase (GSK)-3β, a serine/threonine kinase, regulates multiple cellular processes, such as cell cycle control, differentiation, motility, apoptosis, and inflammation, by modulating numerous transcription factors, including NFAT, NF-κB, AP-1, T-bet, and cyclic adenosine monophosphate (cAMP)-response element-binding protein [9,10]. Most of these transcription factors are important for T lymphocyte activation and cytokine production [5,7]. Principally, GSK-3β is tightly regulated by its phosphorylation at the serine residue 9 (Ser9), which leads to its enzymatic inactivation, and at tyrosine residue 216 (Tyr216), which leads to its activation [11,12]. Several kinases, including p70 S6 kinase, p90 Rsk, cAMP-dependent PKA, Akt (PKB), and PKC, negatively regulate GSK-3β, while proline-rich tyrosine kinase 2 (Pyk2) positively regulates GSK-3β [11,12]. In addition, protein phosphatases (PPs), such as PP1, PP2A, and PP2B, can act as positive regulators of GSK-3β activation by dephosphorylating its serine residue [13,14]. The regulation of GSK-3β and its effects have been widely studied in the context of T lymphocyte activation [9,10]. As a regulator of NFAT nuclear export, inhibiting GSK-3β is speculated to lead to immune enhancement during T lymphocyte responses [15,16]. In this study, by using a combination of the PKC activator 12-O-tetradecanoylphorbol-13-acetate (TPA) and the calcium ionophore ionomycin to activate T lymphocyte responses [17], we investigated the role of the molecular regulation of GSK-3β activation in T lymphocyte-associated cytokine production.

## 2. Materials and Methods

### 2.1. Reagents and Antibodies

Dimethyl sulfoxide (DMSO), 12-O-tetradecanoylphorbol-13-acetate (TPA), ionomycin, GSK-3 inhibitor 6-bromoindirubin-3’-oxime (BIO), and 4’,6-diamidino-2-phenylindole (DAPI) were purchased from Sigma-Aldrich (St. Louis, MO, USA). Mouse monoclonal antibody for β-actin was obtained from Sigma-Aldrich. Antibodies against phospho-GSK-3β at serine 9 (Ser9) or tyrosine 216 (Tyr216), GSK-3β, Pyk2, horseradish peroxidase (HRP)-conjugated horse anti-mouse and HRP-conjugated goat anti-rabbit (Cell Signaling Technology, Beverly, MA, USA); PE-conjugated anti-mouse CD4, Alex Fluor 488-conjugated anti-mouse CD8, and PerCP-Cy5.5-conjugated anti-mouse IFN-γ were obtained from BioLegend (San Diego, CA, U); PE-conjugated anti-human CD4 and fluorescein isothiocyanate-conjugated anti-human CD8 were from BD PharmingenTM (San Diego, CA, USA).

### 2.2. Mice and Thymocyte Isolation

Six-week-old male progeny of wild-type (WT) C57BL/6 mice were fed standard laboratory chow and water ad libitum in the Laboratory Animal Center of National Cheng Kung University. The mice were raised and cared for according to the guidelines set by the Ministry of Science and Technology, Taiwan. The experimental protocol adhered to the rules of the Animal Protection Act of Taiwan and was approved by the Laboratory Animal Care and Use Committee of National Cheng Kung University (IACUC Approval No. 102180). The mice were given a lethal overdose of pentobarbital (200 mg/kg, intraperitoneal), and their thymic tissue was harvested according to previous work [18]. Mouse thymocytes were carefully layered over Histopaque-1083 (Sigma-Aldrich) and centrifuged at 400× *g* for 30 min at 20 °C in a swinging-bucket centrifuge without braking. The upper layer was aspirated, leaving the mononuclear cell layer undisturbed at the interphase, and the cellular layer was transferred to a new tube. RPMI 1640 (Invitrogen Life Technologies, Rockville, MD) cell culture medium was added to the cells, and the samples were centrifuged at 500× *g* for 30 min at 20 °C. The supernatants were carefully removed, and the cellular pellets were resuspended in RPMI 1640.

### 2.3. Human T Lymphocyte Isolation

The collection and analysis of human whole blood samples followed the protocols and procedures of the institutional review board (ER-98-167 and A-ER-102-123) of National Cheng Kung University Hospital (Tainan, Taiwan). First, whole blood samples were diluted in phosphate buffered saline (PBS) containing 0.5% bovine serum albumin and 2 mM ethylenediaminetetraacetic acid (EDTA) with 2–4 volumes of buffer. To isolate the peripheral blood mononuclear cells (PBMCs), the diluted samples were carefully layered over Histopaque-1077 (Sigma-Aldrich) and centrifuged at 400× *g* for 25 min at 20 °C in a swinging-bucket centrifuge without braking. Following PBMC isolation, human CD4^+^ and CD8^+^ T cells were individually separated using the BD IMagTM Human CD4 or CD8 Lymphocyte Enrichment Set-DM kits according to the manufacturer’s instructions (https://www.bdbiosciences.com/us/reagents/research/magnetic-cell-separation/human-cell-separation-reagents/human-cd4-t-lymphocyte-enrichment-set-dm/p/557939).

### 2.4. Cell Culture

Human Jurkat T cells (ATCC, TIB-152), isolated C57BL/6 thymocytes, and isolated human CD4^+^/CD8^+^ T lymphocytes were grown in RPMI 1640 medium (RPMI; Invitrogen Life Technologies, Rockville, MD, USA) with L-glutamine and sodium pyruvate supplemented with 10% heat-inactivated fetal bovine serum (FBS; Invitrogen Life Technologies), 50 U/mL penicillin and 50 µg/mL streptomycin. The cells were maintained in a humidified atmosphere with 5% CO_2_ and 95% air at 37 °C.

### 2.5. Immunostaining

To detect CD4, CD8, and IFN-γ, cells were fixed with 4% paraformaldehyde, blocked, and then incubated with the indicated fluorophore-conjugated primary antibodies at 4 °C for 30 min. The samples were analyzed using flow cytometry (FACSCalibur; BD Biosciences, San Jose, CA, USA) with excitation wavelengths of 488 nm and 633 nm; emission was detected with the FL-1 channel (515–545 nm) and the FL-2 channel (525–625 nm). The samples were analyzed using CellQuest Pro 4.0.2 software (BD Biosciences), and quantification was conducted using the WinMDI 2.8 software (The Scripps Institute, La Jolla, CA, USA).

### 2.6. Enzyme-Linked Immunosorbent Assay (ELISA)

To measure cytokine production, commercial ELISA kits were used to detect the concentrations of human and murine IFN-γ (Invitrogen Corporation.) (https://www.thermofisher.com/elisa/product/IFN-gamma-Mouse-ELISA-Kit/BMS606), TNF-α, and IL-2 (R&D Systems, Minneapolis, MN, USA) (https://www.rndsystems.com/products/human-tnf-alpha-quantikine-elisa-kit_dta00d) in the cell culture medium and murine serum, according to the manufacturer’s instructions. The plates were read at 450 nm on a microplate reader (SpectraMax 340PC; Molecular Devices), and the data were analyzed using the SoftMax Pro software (Molecular Devices).

### 2.7. Cytotoxicity Assay

To evaluate cell damage, lactate dehydrogenase (LDH) activity was assayed using a colorimetric assay (Cytotoxicity Detection kit; Roche Diagnostics, Lewes, UK) according to the manufacturer’s instructions (https://www.sigmaaldrich.com/content/dam/sigma-aldrich/docs/Roche/Bulletin/1/11644793001bul.pdf).

### 2.8. Western Blotting

The harvested cells were lysed by using lysis buffer containing 1% Triton X-100, 50 mM Tris, pH 7.5, 10 mM EDTA, 0.02% NaN_3_, and a protease inhibitor cocktail (Roche Diagnostics, Mannheim, Germany). The proteins were separated by SDS-polyacrylamide gel electrophoresis (SDS-PAGE) and then transferred to polyvinylidene difluoride membranes (Millipore Corporation, Billerica, MA). Following the general online protocols for Western blot analysis (https://www.sigmaaldrich.com/technical-documents/protocols/biology/western-blotting.html), the blots were developed with an enhanced chemiluminescence Western blot detection kit (Millipore Corporation) according to the manufacturer’s instructions (https://www.merckmillipore.com/TW/zh/life-science-research/protein-detection-quantification/western-blotting/protein-detection/chemiluminescent-westerns/E2ab.qB.lLcAAAFBVoIRRkw6,nav). The protein band intensity was analyzed using quantitative autoradiography densitometry and the changes in the ratio of proteins compared with the normalized value of control groups (phosphorylated protein/total protein/β-actin) are also shown.

### 2.9. Short Hairpin RNA (shRNA)

The protein was silenced by using lentiviral expression of short hairpin RNA (shRNA) targeting the human GSK-3β (TRCN0000010551, containing the following shRNA target sequence: 5’-CACTGGTCACGTTTGGAAAGA-3’) and a negative control construct (luciferase shRNA, shLuc). The shRNA clones were obtained from the National RNAi Core Facility, Institute of Molecular Biology/Genomic Research Center, Academia Sinica, Taipei, Taiwan. The lentiviruses were prepared, and the cells were infected according to previously described protocols [19]. Briefly, Jurkat T cells were transduced with a lentivirus at an appropriate multiplicity of infection in complete growth medium supplemented with polybrene (Sigma-Aldrich). After transduction for 24 h and puromycin (Calbiochem, San Diego, CA, USA) selection for six days, protein expression was monitored using Western blot analysis.

### 2.10. Calcineurin Cellular Activity Assay

The calcineurin activity was measured using a calcineurin cellular activity assay kit (Calbiochem) according to the manufacturer’s instructions (https://www.merckmillipore.com/TW/zh/product/Calcineurin-Cellular-Activity-Assay-Kit-Colorimetric,EMD_BIO-207007).

### 2.11. Fluorescence Imaging

The cells were fixed in 3.7% formaldehyde in PBS for 20 min. The cells were washed with PBS twice and then incubated with the primary antibody at 4 °C overnight. The samples were washed with PBS twice before incubation with Alexa Fluor 488-labeled antibody at room temperature for 1 h. DAPI (5 μg/mL, Sigma-Aldrich) was used for nuclear staining and incubated at room temperature for 30 min. The cells were washed with PBS and observed under a fluorescence microscope (BX51; Olympus, Tokyo, Japan). For quantification, three fields of view (100× total magnification) are counted, and the percentages of positive cells are calculated.

### 2.12. RNA Interference

T-bet expression was silenced using commercialized siRNA (TBX21HSS121201, containing the following siRNA target sequences: 5’-GCAACGCUUCCAACACGCAUAUCUU-3’ and 5’-AAGAUAUGCGUGUUGGGAGCGUUGC-3’; Invitrogen). Transfection was performed by electroporation using a pipette-type microporator (Microporator system; Digital Bio Technology, Suwon, Korea). After transfection, the cells were incubated in RPMI for 16 h at 37 °C before treatment. A nonspecific scrambled siRNA kit (StealthTM RNAi Negative Control Duplexes, 12935-100; Invitrogen) was the negative control.

### 2.13. Animal Treatment

To establish murine inflammation, mice were intravenously injected through the tail vein with 50–200 μg/kg TPA and 250 μg/kg ionomycin dissolved in sterile PBS; the concentrations were adjusted for a total volume of 200 μL per injection. To verify the protective effect of GSK-3 inhibition, the mice were pretreated with 2 mg/kg BIO, as referred to in previous work [20], diluted in PBS in a total volume of 200 μL for 0.5 h, as previously described. PBS was used as the vehicle control.

### 2.14. Statistical Analysis

Values from three independent experiments are the mean ± standard deviation (SD). The groups were compared by using Student’s two-tailed unpaired t-test. Statistical significance was set at *p* < 0.05.

## 3. Results

### 3.1. 12-O-Tetradecanoylphorbol-13-Acetate (TPA) and Ionomycin (T/I) Treatment Induces Cytokine Production in Different T Lymphocyte Lineages

A combination treatment of TPA and ionomycin is generally used to mimic TCR signaling to activate T lymphocytes [17]. To investigate the induction of cytokine production in different subsets of TPA-, ionomycin-, or T/I-treated T lymphocytes, fluorescence-activated cell sorting analysis, and ELISA were performed in this study. Among purified human CD4^+^ T lymphocytes, 77.4% of the cells expressed IFN-γ after T/I treatment; however, among purified human CD8^+^ T lymphocytes, only 18.7% of the cells expressed IFN-γ after T/I treatment (Figure 1A). A quantitative ELISA confirmed that, compared with TPA or ionomycin treatment, only T/I treatment significantly (*p* < 0.001) induced IFN-γ production, mainly in CD4^+^ T lymphocytes (Figure 1B). Next, we confirmed the findings that only T/I treatment caused significant (*p* < 0.001) IFN-γ production in Jurkat T cells at an early time point, namely, 6 h (Figure 1C). Furthermore, a flow cytometry-based analysis of murine C57BL/6 thymocytes demonstrated that T/I treatment induced IFN-γ expression in both CD4^+^CD8^−^ (50.1%) and CD4^+^CD8^+^ (72.6%) thymocytes but not in CD4^−^CD8^+^ (2.6%) thymocytes (Figure 1D). These results suggest that T/I treatment induces IFN-γ production primarily in CD4^+^ T lymphocytes.

To confirm the regulation of cytokine induction by TPA and ionomycin, purified human CD4^+^ T lymphocytes were next selected in this study. Following TPA, ionomycin, or T/I treatment without cytotoxicity, the ELISA results showed that only T/I treatment significantly (*p* < 0.001) induced the production of several T cell-associated cytokines, including IFN-γ, TNF-α, and IL-2, in three individual volunteers of number (No.) 1 to 3 (Figure 1E). As TPA is known as a protein kinase C (PKC) activator and ionomycin is a calcium ionophore that increases the intracellular calcium concentration [17], pharmacological targeting was performed in this study. Using a broad PKC inhibitor, bisindolylmaleimide (Bis), or a calcium chelator, 1,2-bis(o-aminophenoxy)ethane-*N,N,N′,N*′-tetraacetic acid (BAPTA), the results showed that inhibiting PKC and calcium release significantly (*p* < 0.001) decreased the T/I-induced IFN-γ, TNF-α, and IL-2 production in CD4^+^ T lymphocytes without further cytotoxic response (Figure 1F). These results demonstrate that PKC activation and calcium influx are both important for T/I-mediated cytokine production in CD4^+^ T lymphocytes.

### 3.2. GSK-3β Regulates T/I-activated Cytokine Production

The serine/threonine kinase GSK-3β is able to modulate T lymphocyte activation in response to TCR, cytokine, and growth factor stimulation [21,22]. However, the mechanism of GSK-3β regulation during T/I-induced T lymphocyte activation is undefined. Western blotting showed that T/I treatment caused temporal GSK-3β activation, which was characterized by dephosphorylation at serine 9 (Ser9) (0.5 to 3 h posttreatment) accompanied by phosphorylation at tyrosine 216 (Tyr216) (0.5 to 12 h posttreatment) (Figure 2A). Without causing cytotoxicity, pharmacological treatment with the GSK-3 inhibitor BIO significantly (*p* < 0.01) decreased the T/I-induced production of IFN-γ, TNF-α, and IL-2 in three individual volunteers of CD4^+^ T lymphocytes and Jurkat T cells (Figure 2B). To further evaluate the role of GSK-3β in T/I-activated cytokine production, lentiviral-based short hairpin RNA (shRNA) was used to knockdown GSK-3β in Jurkat T cells (Figure 2C). The knockdown of GSK-3β significantly (*p* < 0.01) decreased T/I-induced IFN-γ, TNF-α, and IL-2 production (Figure 2D). These findings indicate that GSK-3β positively regulates T/I-induced T cell-associated cytokine production.

### 3.3. T/I Treatment Induces GSK-3β-Mediated T Cell-Associated Cytokine Production through the Pyk2-Regulated Pathway

The tyrosine kinase Pyk2 positively regulates GSK-3β by causing its phosphorylation at tyrosine 216 (Tyr216) [23]. Notably, Pyk2 activation is also regulated by the PKC and calcium/calmodulin signaling pathways [24]. The pharmacological inhibition of Pyk2 by using Tyrphostin A9 effectively decreased the T/I-induced GSK-3β phosphorylation at Tyr216 (Figure 3A). The blockade of Pyk2 also significantly (*p* < 0.001) reduced the T/I-induced IFN-γ, TNF-α, and IL-2 production in CD4^+^ T lymphocytes and Jurkat T cells without causing cytotoxicity (Figure 3B). These data demonstrate that Pyk2 is a positive regulator of T/I-induced GSK-3β activation, which coordinately regulates cytokine production in T lymphocytes.

### 3.4. PP2B Regulates T/I-Induced GSK-3β Activation and Cytokine Production

In addition to Pyk2-regulated GSK-3β activation, PP2B (also known as calcineurin) is a calcium-dependent phosphatase that dephosphorylates serine 9 (Ser9) of GSK-3β, ultimately leading to GSK-3β activation [25]. During T/I stimulation, treatment with the PP2B inhibitor cyclosporine A effectively increased GSK-3β phosphorylation at Ser9 (Figure 4A). Regarding the involvement of the PKC and calcium signaling pathways in controlling PP2B [16], treatment with the PKC inhibitor Bis and the calcium inhibitor BAPTA significantly (*p* < 0.001) inhibited T/I-activated PP2B at 1 h poststimulation (Figure 4B). Intracellular signal pathways, including PP2B and mitogen-activated protein kinases (MAPKs), transmit signals to activate transcription factors, such as NFAT and NF-κB, to activate T lymphocytes [16,26]. In T/I-treated CD4^+^ T lymphocytes, the blockade of PP2B significantly (*p* < 0.001) inhibited IFN-γ, TNF-α, and IL-2 production. Notably, treating cells with the NFAT inhibitor peptide Met-Ala-Gly-Pro-His-Pro-Val-Ile-Val-Ile-Thr-Gly-Pro-His-Glu-Glu (VIVIT), the NF-κB inhibitor caffeic acid phenethyl ester (CAPE) (Figure 4C), and the MAPK inhibitor PD98059 (Figure 4D) did not attenuate T/I-stimulated cytokine production. Additionally, treatment with combined inhibitors, including VIVIT, CAPE, and PD98059, did not inhibit T/I-stimulated IFN-γ production in CD4^+^ T lymphocytes (Figure 4E). These results indicate that T/I stimulation induces PP2B activation, leading to GSK-3β-regulated cytokine production.

### 3.5. T/I Treatment Induces GSK-3β-Regulated T-bet Nuclear Translocation Followed by T-bet-Mediated Cytokine Production

Regulation of GSK-3β is able to modulate a variety of transcription factors, including NFAT, NF-κB, CREB, AP-1, and T-bet, to control T lymphocyte activation [22]. Our studies excluded the involvement of NFAT, NF-κB, and extracellular signal-regulated kinase-associated cAMP response element binding (CREB) and AP-1 during T/I-induced T lymphocyte activation. Accordingly, to control IFN-γ, TNF-α, and IL-2 production for type 1 helper T (Th1) differentiation, the transcription factor T-bet is also critical [27]. To investigate whether T-bet facilitates IFN-γ, TNF-α, and IL-2 production in T/I-activated CD4^+^ T lymphocytes, fluorescent immunostaining was used. The results showed that T/I stimulation significantly (*p* < 0.001) induced T-bet nuclear translocation at 6 h poststimulation, while treatment with the GSK-3 inhibitor BIO significantly (*p* < 0.001) blocked this effect (Figure 5A). To further evaluate the role of T-bet, small interfering RNA (siRNA) was used to silence the expression of T-bet in Jurkat T cells. The knockdown of T-bet significantly (*p* < 0.001) reduced T/I-stimulated IFN-γ, TNF-α, and IL-2 production (Figure 5B). These findings indicate that T/I treatment induces GSK-3β-regulated T-bet nuclear translocation and T-bet-regulated cytokine production.

### 3.6. Pharmacological Inhibition of GSK-3 Reduces Mortality and Suppresses Cytokine production in T/I-Treated Mice

To investigate whether GSK-3β regulates T lymphocyte activation in vivo, without the supporting protocols created by previous works, we generated an in vivo model in which C57BL/6 mice were intravenously injected with different doses of T/I. The survival rate showed that T/I (200 μg/kg TPA plus 250 μg/kg ionomycin) significantly (*p* < 0.001) increased mouse death 6 to 12 h posttreatment. Notably, treatment with the GSK-3 inhibitor BIO (2 mg/kg) significantly (*p* < 0.001) increased the survival of the T/I-stimulated mice (Figure 6A). A quantitative ELISA showed that inhibiting GSK-3β significantly (*p* < 0.001) suppressed T/I-stimulated increases in the expression of serum IFN-γ (Figure 6B) and TNF-α (Figure 6C) in the mice. The data suggest that blockade of GSK-3β prevents T/I-mediated mortality and represses T cell-associated cytokine production in mice.

## 4. Discussion

In response to T/I stimulation, which is used to mimic TCR signal transduction, activated CD4^+^ T lymphocytes secrete a variety of cytokines, including IFN-γ, TNF-α, and IL-2, to regulate T lymphocyte responses. As the TPA-activated PKC and ionomycin-induced calcium influx signaling pathways are speculated to mediate T lymphocyte activation, these pathways are essential for T/I-stimulated cytokine production. The findings of this study, as summarized in Figure 6D, demonstrate that T/I further induces the activation of PP2B and Pyk2 to cause the dephosphorylation of GSK-3β (Ser9) and the phosphorylation of GSK-3β (Tyr216), respectively, which are required to activate GSK-3β [23,25]. Following GSK-3β-mediated T-bet nuclear translocation, our findings illustrate that the essential PKC/calcium/Pyk2/PP2B/GSK-3β/T-bet signaling axis is involved in T/I-induced cytokine production in CD4^+^ T lymphocytes. Our further in vivo study, in which GSK-3β was pharmacologically inhibited, confirms the therapeutic efficacy of inhibiting GSK-3β against T/I-induced mortality and cytokine production in mice. In addition to exploring the molecular basis of GSK-3β regulation in T/I-activated CD4^+^ T lymphocytes, this study also provides a potential immunosuppressive therapy involving the targeting of GSK-3β.

To create experimental models of CD4^+^ T lymphocyte activation in vitro and in vivo, a combination of TPA and ionomycin, the so-called T/I model, was utilized in this work [17]. Consistent with TCR signaling [7], the activation of PKC and the influx of calcium, which lead to PP2B activation, are essential for regulating cytokine production. However, inconsistent with TCR signaling, the GSK-3β, and NFAT signaling pathways are unexpectedly deregulated in the T/I model of CD4^+^ T lymphocyte activation. As shown in this study, the supporting data indicate the GSK-3β-dependent but NFAT-independent regulation of T/I-induced cytokine responses. In contrast, during TCR signaling, active GSK-3β is able to export nuclear NFAT by inhibiting its DNA binding activity from downregulating NFAT-regulated immune modulation [16,26,28,29]. Inhibitors of GSK-3β lead to the enhancement of NFAT-regulated immune cytokine responses in T lymphocytes [10,29]. The signaling pathways of the TCR and T/I, which mimics TCR, share PKC/calcium influx/PP2B axis signaling but could be different downstream of GSK-3β signaling. Accordingly, it is speculated that the regulation of GSK-3β is vital for T lymphocyte function, and the dual role of GSK-3β may be exerted mainly by its dynamic activation and inactivation, which are regulated by multiple signaling pathways during T lymphocyte activation.

The regulation of GSK-3β is able to modulate several transcription factors, including NFAT, NF-κB, AP-1, and T-bet, which are critical in T lymphocyte responses, such as the cell cycle, proliferation, differentiation, motility, and inflammation [10]. We found that T/I stimulation induces Pyk2- and PP2B-mediated GSK-3β activation, followed by GSK-3β-regulated T-bet nuclear translocation and T-bet-mediated IFN-γ, TNF-α, and IL-2 production. It is well known that T/I stimulation causes the activation of NFAT, NF-κB, AP-1, and T-bet to control T lymphocyte-associated cytokine production [30,31,32,33,34]. However, in T/I-activated CD4^+^ T lymphocytes, the pharmacological inhibition of NFAT, NF-κB, and MAPK-activated AP-1 is unable to suppress cytokine production. Alternatively, GSK-3β-regulated T-bet is indispensable for triggering such effects. T-bet is a member of the T-box family of transcription factors that can bind to the promoters of IFN-γ and TNF-α to regulate Th1 cytokine expression [27,35,36].

CD4^+^ T lymphocytes differentiate into several lineages with distinct effector cellular functions, such as Th1 cells that stably secrete IFN-γ, Th2 cells that express IL-4, Th17 cells that produce IL-17, and iTreg cells that generate IL-10 and TGF-β [37]. To achieve the plasticity and flexibility of CD4^+^ T cell differentiation, the specific ‘‘master’’ transcription factors determine cellular differentiation; T-bet mediates Th1 cell differentiation, GATA3 mediates Th2 cell differentiation, retinoid-related orphan receptor γt mediates Th17 cell differentiation, and Foxp3 mediates iTreg cell differentiation. According to this study, T/I treatment stimulates CD4^+^ T lymphocytes to secrete IFN-γ, TNF-α, and IL-2. These results suggest that T/I-activated CD4^+^ T lymphocytes may differentiate into Th1 cells, while the blockade of GSK-3β inhibits these effects. T/I stimulation usually leads to T lymphocyte activation rather than cell differentiation. The immune modulation by T/I combination treatment through the regulation of T lymphocyte differentiation needs further investigation.

In this study, by using an in vivo model of T/I-induced cytokine production and mortality, the pharmacological inhibition of GSK-3β was shown to protect against these effects. The results of this study correlate with previous findings that GSK-3β activation has been found to be significant in mice with allograft rejection diseases, such as graft-versus-host disease (GVHD) [38]. Pharmacologically targeting PP2B by using cyclosporine A and FK506 is widely utilized in clinical treatments [39]. While targeting NFAT also shows potent therapeutic efficacy against GVHD [40], its clinical efficacy and PP2B-NFAT or PP2B-GSK-3β activation in CD4^+^ lymphocytes of GVHD patients need to be validated. As demonstrated in the T/I model of CD4^+^ lymphocyte activation, inhibiting PP2B and GSK-3β, but not NFAT effectively suppresses cytokine production. To mimic GVHD, the immunological features of immune activation and tissue/organ damage in T/I-stimulated mice need further validation.

In conclusion, this study is the first to demonstrate a model of GSK-3β-mediated T lymphocyte activation via the PKC/calcium/Pyk2/PP2B/GSK-3β/T-bet signaling axis in T/I-activated CD4^+^ lymphocytes. In the model of T/I-activated CD4^+^ lymphocytes, neither the NFAT nor NF-κB signaling pathway is required for cytokine production. Additionally, MAPKs are not required for the T/I-induced activation of CD4^+^ lymphocytes. As our results show an essential role for GSK-3β-mediated T-bet signaling in the T/I-induced activation of CD4^+^ lymphocytes, further research of the identified signaling pathway will lead to a strategy for preventing abnormal T lymphocyte activation and may lead to treatments for diseases such as GVHD. Our previous works demonstrated the blockade of GSK-3 against TPA-induced skin inflammation [19]. However, TPA treatment may also induce GSK-3 inactivation in fibroblast cells [41]. In this study, the T/I combination is only checked for GSK-3 activation. During T cell activation, the dual regulation for modulating GSK-3 activation needs further examination. Additionally, this study needs to be extended by using antigen-specific stimulation of CD4^+^ lymphocytes in vitro and in vivo.

## Figures and Tables

**Figure 1 cells-09-01424-f001:**
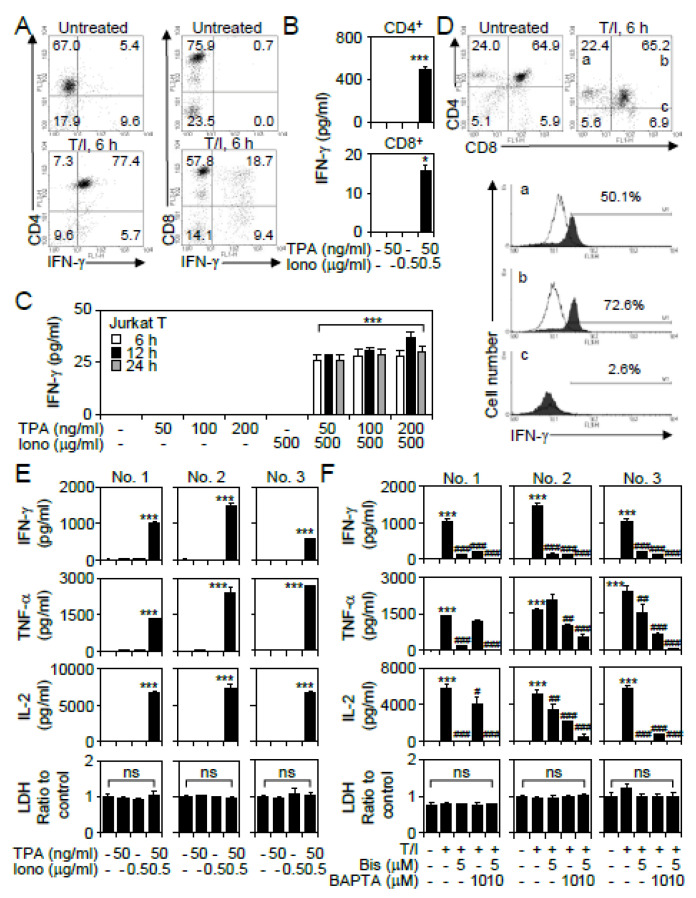
A combination of 12-O-tetradecanoylphorbol-13-acetate (TPA) and ionomycin causes in vitro activation of CD4^+^ T lymphocytes. (**A**) Representative dot plots of immunostaining followed by flow cytometric analysis detected the IFN-γ expression in CD4^+^ and CD8^+^ T lymphocytes 6 h posttreatment with TPA plus ionomycin (T/I). In addition, ELISA determined the levels of IFN-γ in the cell supernatants of TPA-, ionomycin (Iono)-, and T/I-treated CD4^+^ and CD8^+^ T lymphocytes (**B**) and Jurkat T cells (**C**). (**D**) Representative dot plots and histograms of immunostaining followed by flow cytometric analysis detected the IFN-γ expression in T/I-stimulated murine CD4^+^CD8^−^ (a), CD4^+^CD8^+^ (b), and CD4^−^CD8^+^ (c) thymocytes. Primary CD4^+^ T cells were obtained from three individual volunteers of number (No.) 1 to 3. ELISA determined the levels of cytokines in the cell supernatants of TPA-, ionomycin-, or T/I-treated CD4^+^ T cells 6 h posttreatment (**E**) without or (**F**) with Bis (5 μM) and BAPTA (10 μM) pretreatment for 0.5 h. An LDH assay was used to detect cell cytotoxicity, and the results are normalized to the untreated group. For flow cytometric analysis, the percentages of IFN-γ-expressing cells are shown. For ELISA, the data are shown as the mean ± SD from three individual experiments. * *p* < 0.05 and *** *p* < 0.001 compared to untreated cells. # *p* < 0.05, ## *p* < 0.01, and ### *p* < 0.001 compared to the T/I-treated group. ns, not significant.

**Figure 2 cells-09-01424-f002:**
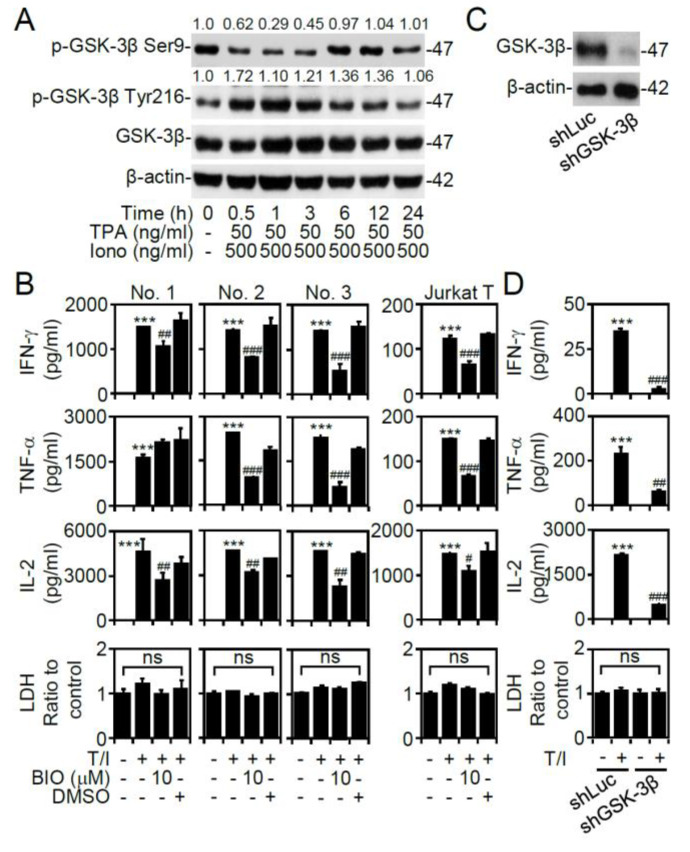
Treatment with T/I activates GSK-3β and then induces GSK-3β-regulated cytokine production. (**A**) Western blotting detected the phosphorylation of GSK-3β (47 kDa) in CD4^+^ lymphocytes treated with the T/I combination. (**B**) Primary CD4^+^ T cells were obtained from three individual volunteers of number (No.) 1 to 3. ELISA determined the levels of cytokines in the cell supernatants of T/I-treated CD4^+^ T lymphocytes and Jurkat T cells 6 h posttreatment after BIO (10 μM) pretreatment for 0.5 h. DMSO was used as a control. (**C**) Western blot analysis showed GSK-3β (47 kDa) knockdown in Jurkat T cells (shGSK-3β). Luciferase shRNA (shLuc) was used as a control. (**D**) ELISA determined the levels of cytokines in the shGSK-3β-transfected cells pretreated with BIO (10 μM) for 0.5 h and then stimulated with or without T/I for an additional 6 h. For Western blotting, β-actin (42 kDa) was used as an internal control. The ratio of phosphorylated protein/total protein/β-actin is shown. For ELISA, an LDH assay was used to detect cell cytotoxicity, and the results are normalized to the untreated group. The data are shown as the mean ± SD from three individual experiments. *** *p* < 0.001 compared to the untreated cells. # *p* < 0.05, ## *p* < 0.01, and ### *p* < 0.001 compared to the T/I-treated group. ns, not significant.

**Figure 3 cells-09-01424-f003:**
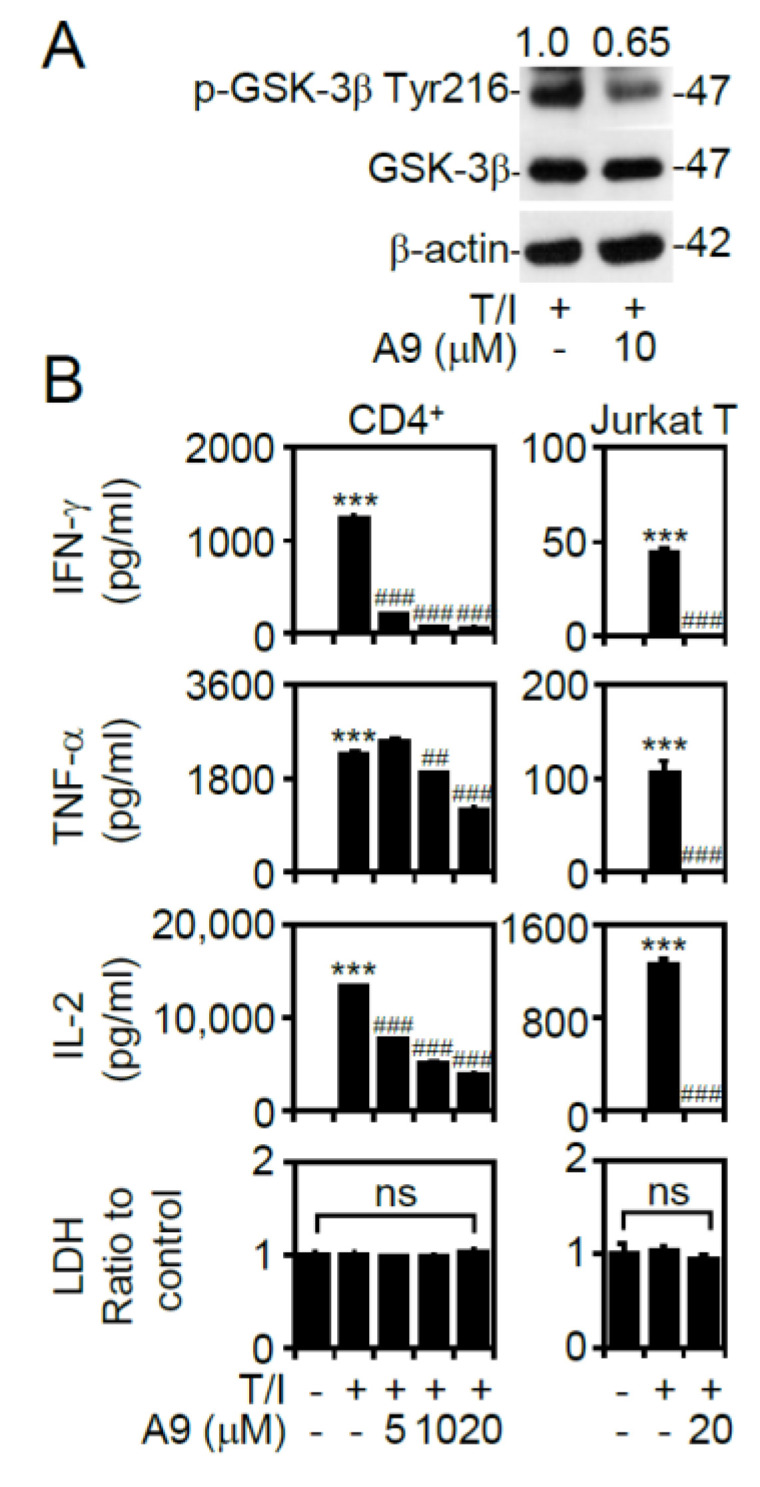
Treatment with T/I induces Pyk-2-regulated GSK-3β activation and cytokine production. (**A**) Western blotting detected the phosphorylation of GSK-3β (47 kDa) in CD4^+^ lymphocytes treated with the T/I combination in the presence of A9 (10 μM). (**B**) ELISA determined the levels of cytokines in the cell supernatants of T/I-treated CD4^+^ T lymphocytes and Jurkat T cells 6 h posttreatment after A9 pretreatment for 0.5 h. For Western blotting, β-actin (42 kDa) was used as an internal control. The ratio of phosphorylated protein/total protein/β-actin is shown. For ELISA, an LDH assay was used to detect cell cytotoxicity, and the results are normalized to the untreated group. The data are shown as the mean ± SD from three individual experiments. *** *p* < 0.001 compared to the untreated cells. ## *p* < 0.01 and ### *p* < 0.001 compared to the T/I-treated group. ns, not significant.

**Figure 4 cells-09-01424-f004:**
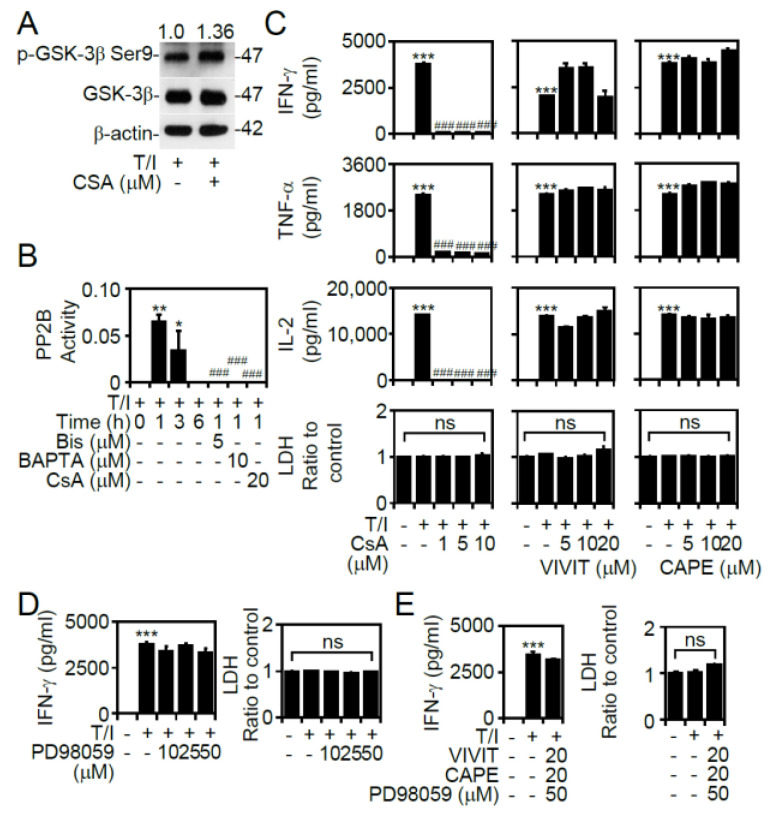
Treatment with T/I induces GSK-3β activation and cytokine production through calcineurin (PP2B) signaling. (**A**) Western blotting detected the phosphorylation of GSK-3β (47 kDa) in CD4^+^ lymphocytes treated with the T/I combination in the presence of cyclosporine A (CsA, 10 μM). CD4^+^ lymphocytes were pretreated with Bis, BAPTA, CsA, VIVIT, CAPE, PD98059, or a combination of VIVIT, CAPE, and PD98059 for 0.5 h and then stimulated with T/I for 6 h. (**B**) The PP2B activity assay detected the phosphatase activity. (**C**, **D**, and **E**) ELISA determined the levels of cytokines in the cell supernatants of the T/I-treated cells. For Western blotting, β-actin (42 kDa) was used as an internal control. The ratio of phosphorylated protein/total protein/β-actin is shown. For ELISA, an LDH assay was used to detect cell cytotoxicity, and the results are normalized to the untreated group. The data are shown as the mean ± SD from three individual experiments. * *p* < 0.05, ** *p* < 0.01, and *** *p* < 0.001 compared to the untreated cells. ### *p* < 0.001 compared to the T/I-treated group. ns, not significant.

**Figure 5 cells-09-01424-f005:**
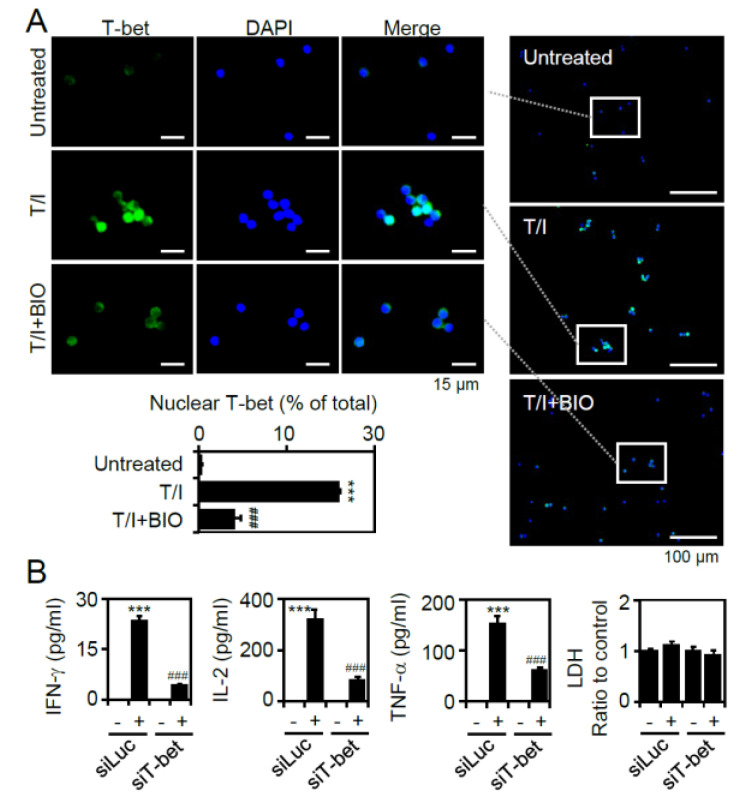
Pharmacological inhibition of GSK-3β decreases the nuclear translocation of T-bet, which is required for controlling T lymphocyte cytokines. (**A**) Representative selected images of T-bet (*green*) immunostaining, which are obtained from the field of view (100× total magnification), in Jurkat T cells pretreated with BIO for 0.5 h and then stimulated with T/I for an additional 6 h. DAPI (*blue*) was used for nuclear staining. The percentages of positive cells, which were calculated from three fields of view (100× total magnification), are shown. (**B**) ELISA determined the levels of cytokines in the cell supernatants of T/I-treated Jurkat T cells transfected with siRNA targeting T-bet (siT-bet). siRNA targeting luciferase (siLuc) was used as a nonspecific control. For ELISA, an LDH assay was used to detect cell cytotoxicity, and the results are normalized to the untreated group. The data are shown as the mean ± SD from three individual experiments. *** *p* < 0.001 compared to the untreated cells. ### *p* < 0.001 compared to the T/I-treated group. ns, not significant.

**Figure 6 cells-09-01424-f006:**
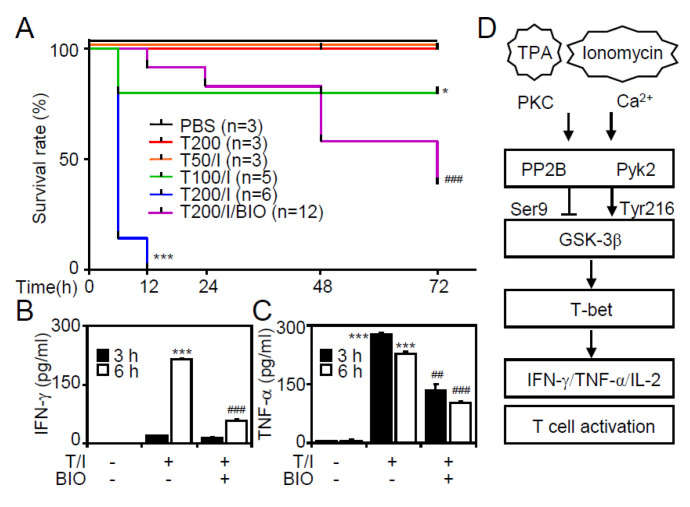
Treatment with a GSK-3β inhibitor reduces T/I-induced mortality and cytokine production in vivo. (**A**) Survival rate of mice following T/I treatment with or without BIO therapy. Six groups were included: PBS (n = 3 for control), T200 (n = 3; TPA, 200 μg/kg), T50/I (n = 3; TPA, 50 μg/kg plus ionomycin, 250 μg/kg), T100/I (n = 5; TPA, 100 μg/kg plus ionomycin, 250 μg/kg), T200/I (n = 6; TPA, 200 μg/kg plus ionomycin, 250 μg/kg), and T200/I/BIO (n = 12; TPA, 200 μg/kg plus ionomycin, 250 μg/kg plus BIO 2 mg/kg). (**B** and **C**) Mice were pretreated with BIO (2 mg/kg) for 0.5 h and were then stimulated with T/I (n = 3; TPA, 200 μg/kg plus ionomycin, 250 μg/kg) for the indicated time. ELISA determined the levels of IFN-γ and TNF-α in the mouse serum. The data are shown as the mean ± SD obtained from three mice. * *p* < 0.05 and *** *p* < 0.001 compared with the PBS-treated group. ## *p* < 0.01 and ### *p* < 0.001 compared with the T/I-treated group. (**D**) A model of GSK-3β-regulated cytokine production in T/I-activated human CD4^+^ T lymphocytes.

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
