# Peer review of "Glycogen Synthase Kinase-3β Facilitates Cytokine Production in 12-O-Tetradecanoylphorbol-13-Acetate/Ionomycin-Activated Human CD4+ T Lymphocytes"

_cells, 2020, doi:10.3390/cells9061424_

Round 1
Reviewer 1 Report
Tsuai et l present an interesting study on the possible role of GSK-3beta in facilitating cytokine production in T/I activated CD4 T cells. Although some of the data is quite compelling I think the data is patchy in places and needs strengthening.
The authors use a combined treatment of TPA and ionomycin to mimic CD4 T cell activation. This activation is shown in figure 1, and demonstrates that this involves PKC activation and calcium influx.
Figure 2 looks at the effect of T/I on GSK-3beta phosphorylation. The authors show nicely the dephosphorylation of GSK-3 between 0.5h and 3 h. The combined treatment is necessary for CD4 activation as shown in fig 1, but is it necessary for effects on GSK-3? Controls for each treatment alone should also be shown. Also the magnitude of IFNg production and IL-2 is far less with T/I than in any other figure. The IFNg is 5 fold lower than other figures and although BIO does have an effect on this, it does question why the levels were not much higher in the first place.
The paper relies heavily on cytokine production assays until the final two figures, which raise the readers interest. The authors state that inhibition of GSK-3b decreases nuclear translocation of T-bet and although I would agree the data shows this, the field of view is very small. This is looking at jurkat T cells which grow at an extremely high rate, so why only show 4 cells in the control and 5 in the bio treated, even the T/I treatment with 9 cells seem very low. The graph below indicates the percent of cells, this would be better as the actual number of cells. If this reflects the data seen percentages when looking at 4/5 cells could be very misleading. More cells need to shown and the actual number of cells plotted.
Figure 5b, why is the addition of Bio not incoroporated here?
Throughout the paper it is unclear why Bio is used or SH-GSK-3, why not use both, I assume one is to support the other yet they are not used in the same assays. Also why BIO? there are many inhibitors available, none of which have been 100% proven to be effective against only 1 isoform of GSK-3, therefore the data using the sh-GSK-3b is stronger but underused.
Figure 6, the survival plot, I'm not sure I follow this figure. each step should equate to 1 mouse, if the step is double the size it is two mice and so on. so why if there is an n value of 7 for the T200/I/BIO after what looks to be 6 mice deceased is there around 30-40% of mice surviving? T100/I has n=3, yet one step which I would equate to one mouse only drops by about 25%... This does not make any sense.
Minor comments - Figure 2B, the lower panel for LDH in Jurkat cells, please check the labeling of whether BIO or DMSO is used, they seem to be the wrong way around.
Reviewer 2 Report
The manuscript by Tsai and coworkers examined the role of GSK-3beta in CD4+ T lymphocyte cytokine production activated by 12-O-Tetradecanoylphorbol-13-Acetate/Ionomycin. The experiments are well designed and provide evidence that 12-O-Tetradecanoylphorbol-13-Acetate/Ionomycin activates CD4+ lymphocytes by activating GSK3beta via decreasing its phosphorylation on Ser9 and increasing its phosphorylation on Tyr216. They provide further evidence that the increase in Tyr216 phosphorylation is mediated by activated Pyk2, and the decrease in Ser9 phosphorylation is mediated by PP2B. They provide evidence that activated GSK-3beta activates cytokine production by regulating T-bet nuclear translocation. Overall the manuscript is well prepared and the arguments convincing.
A major concern is the reproducibility of some of the specific experiments. Figures 1E, 1F and 2B provided No 1, No 2, and No 3 without explaining what they are. I assume that these are repeated experiments. If that is the case, some of the repeat experiments displayed dramatically different results, such as TNF-a and IL-2 production in 1F. These experiments need to be repeated until there is sufficient confidence that the results are reliable.
There are a few other minor issues that should be taken care of by minor revisions.
1) The manuscript does not provide sufficient references to many statements in the introduction, and experimental methods.
2) The Materials and Methods in general are often too brief, and do not fully explain how the experiments are conducted. As I mentioned earlier, no references are provided for the procedures.
3) Figure 2A provided some numbers above the WB bands. It should be clearly noted what these numbers are. I assume that these are densitometry scan results. If that is correct, the numbers don't seem to correlate well to the bands below. For example, the third p-GSK3b Tyr216 band seems stronger than the second band, but the numbers above are 1.10 for the third band and 1.72 for the second band. Is there some explanation for this?
Round 2
Reviewer 1 Report
Most of the suggested changes have been addressed by the authors and the paper has been strengthened. I would consider the option of including the picture they show in their response regarding figure 5a. There still looks to be few cells considering they are jurkat cells, but the addition of the full figure showing the cut out area would strengthen this somewhat. Otherwise I think the manuscript has improved.
